# Do Instance Priors Help Weakly Supervised Semantic Segmentation?

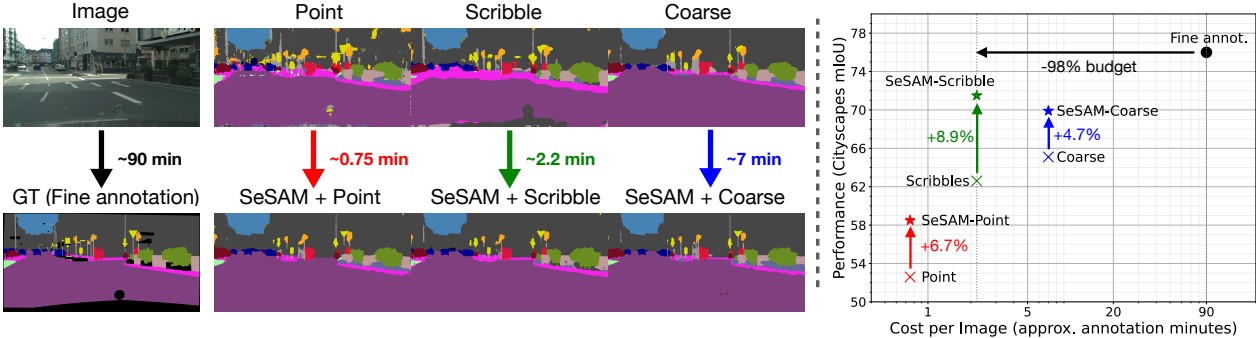

Figure 1: **Our SeSAM framework**, utilizing additional instance priors, integrates with various weak labels (points, scribbles, and coarse annotations) and enhances semantic segmentation quality (see road and car on the left) while remaining cost-effective (right plot). For instance, using scribbles achieves **94%** of fine-supervised performance while requiring only **2%** of the annotation budget.

## Abstract

Semantic segmentation requires dense pixel-level annotations, which are costly and time-consuming to acquire. To address this, we present SeSAM, a framework that uses a foundational segmentation model, i.e. Segment Anything Model (SAM), with weak labels, including coarse masks, scribbles, and points. SAM, originally designed for instance-based segmentation, cannot be directly used for semantic segmentation tasks. In this work, we identify specific challenges faced by SAM and determine appropriate components to adapt it for class-based segmentation using weak labels. Specifically, SeSAM decomposes class masks into connected components, samples point prompts along object skeletons, selects SAM masks using weak-label coverage, and iteratively refines labels using pseudo-labels, enabling SAM-generated masks to be effectively used for semantic segmentation. Integrated with a semi-supervised learning framework, SeSAM balances ground-truth labels, SAM-based pseudo-labels, and high-confidence pseudo-labels, significantly improving segmentation quality. Extensive experiments across multiple benchmarks and weak annotation types show that SeSAM consistently outperforms weakly supervised baselines while substantially reducing annotation cost relative to fine supervision.

## 1 Introduction

Semantic segmentation, the task of assigning semantic labels to each pixel in an image, typically relies on dense pixel-level annotations, which are expensive and time-consuming to collect at scale (Cordts et al., 2016; Neuhold et al., 2017; Zhou et al., 2017). To reduce the annotation effort, foundation models for segmentation, particularly the instance-based SAM (Segment Anything Model) (Kirillov et al., 2023), may appear to offer a universal solution. However, the general-purpose nature of these models limits their direct applicability across different segmentation tasks (VS et al., 2024; Kweon & Yoon, 2024), requiring specific adaptations (Zhang et al., 2023). For instance, Semantic-SAM (Li et al., 2023) proposes complete retraining

of the model to accommodate semantic segmentation. In this work, we present an approach to adapt a foundational instance segmentation model to semantic segmentation, a task that requires pixel-level semantic labels, using weak annotations, including coarse, scribble, and point labels. Our method demonstrates that with careful adaptation, instance priors from foundation models can be effectively leveraged for high-quality semantic segmentation with cost-efficient weak labels, significantly reducing the annotation burden.

While general purpose foundational segmentation models like SAM are powerful for class-agnostic segmentation, applying them to weakly supervised semantic segmentation is non-trivial. SAM is designed to segment *individual* instances, whereas semantic segmentation requires class-wise labels that may span multiple disconnected instances across various locations in an image. Moreover, selecting points as prompts for best segmentation result remains a challenge. SAM also produces masks of varying granularity for the same prompt, often merging multiple semantic objects into a single mask. Since different applications require different levels of segmentation detail, controllable segmentation granularity is essential. To adapt this class-agnostic segmentation model for efficient weakly supervised semantic segmentation (WSSS), our framework addresses each of these challenges, refining SAM's outputs for precise, class-based segmentation tasks[1].

In particular, we introduce SeSAM, a WSSS framework for high-quality semantic segmentation that adapts SAM, a foundation model for instance segmentation, in three steps by answering the following questions: I. *How to use class-wise weak labels to prompt SAM?* II. *How to select points to prompt SAM?* III. *How to choose the relevant masks from SAM?* First, since semantic segmentation is instance-agnostic, we begin by isolating potential instances through morphological operations on coarse segmentation masks. Second, we investigate effective point sampling strategies for prompting the SAM model and selecting the desired segmentation granularity. In particular, we find that equally distributed probability-based sampling along the skeleton of the input shape produces the best quality masks. Third, to resolve the ambiguity in selecting the mask of desired granularity, we propose a selection criterion based on intersection with the weak labels. Additionally, we integrate weak ground-truth segmentation labels with instance priors generated by SAM in a semi-supervised framework, balancing three types of labels: ground-truth labels, SAM-based instance priors, and simple pseudo-labels. To enhance mask quality, we further iteratively refine the weak ground-truth segmentation labels by sampling point prompts from the refined mask. Finally, we show that SeSAM achieves high-quality semantic segmentation with significantly reduced annotation costs compared to fully supervised fine-grained segmentation.

We summarize our main contributions as follows:

- We show that a direct application of foundation instance segmentation models alongside weak annotations performs poorly for semantic segmentation; to address this gap, we propose a three-step pipeline to adapt instance priors from these foundation models to the semantic segmentation task.

- We introduce a semi-supervised framework that iteratively refines the instance priors while balancing between three types of annotations: ground-truth weak semantic segmentation labels, instance priors, and pseudo-labels.

- We provide cost-performance trade-off analysis across different weak annotation types, including point, scribbles, and coarse labels, enabling practitioners to make informed choices on annotation types. In particular, we observe that using SeSAM with scribbles achieves 94% of full supervision performance with only 2% annotation budget.

- We extend WSSS evaluation beyond the commonly used PASCAL (Everingham et al., 2010) benchmark to more challenging Cityscapes (Cordts et al., 2016) and ADE20k (Zhou et al., 2017) datasets. We demonstrate that SeSAM generalizes effectively across multiple weak annotation types and consistently outperforms prior weakly supervised baselines.

---

[1]SAM has been trained with large amounts of annotated segmentation masks. The goal of this paper is to explore how SAM can be leveraged to reduce the annotation effort for semantic segmentation. As we report in the experimental section, the annotation cost for semantic segmentation can be significantly reduced, indicating that annotation costs of SAM can be entirely amortized when systematically used for novel semantic segmentation tasks.

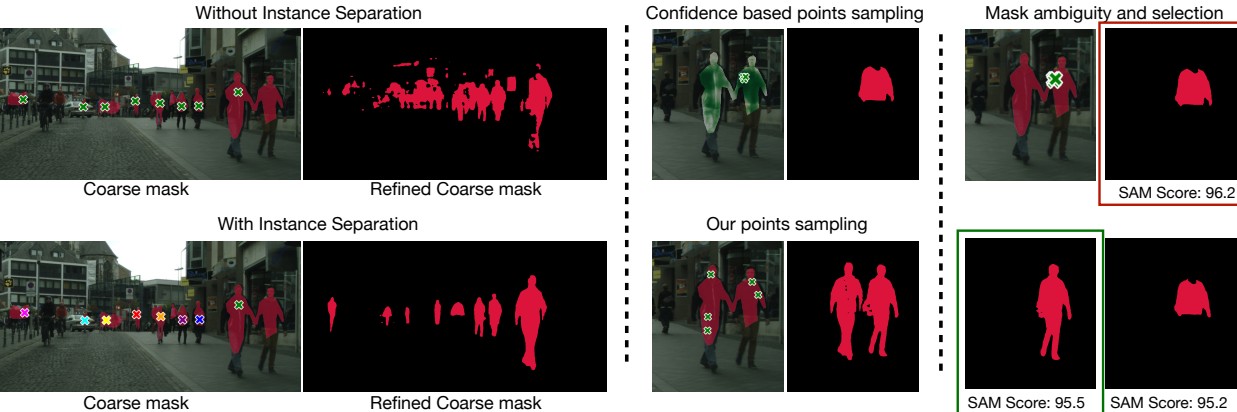

Figure 2: **SAM challenges. Left**: As an instance-based model, SAM performs poorly when segmenting a class comprising multiple instances, see the class "person". Splitting the class mask into individual instances addresses this challenge. **Middle**: Confidence-based sampling (Kweon & Yoon, 2024) yields a suboptimal mask; in contrast, our proposed sampling strategy achieves comprehensive mask coverage, resulting in improved segmentation. **Right**: SAM score-based mask selection may not give the best desired segmentation (see mask in red box). Our mask selection strategy using intersection with the coarse mask determines the desired mask granularity (see mask in green box).

## 2 Related Work

**Weakly Supervised Semantic Segmentation.**

Weakly supervised semantic segmentation (WSSS) aims to reduce annotation effort by leveraging various forms of weak supervision, including image-level labels (Pathak et al., 2015; Wei et al., 2017; Ahn & Kwak, 2018; Sun et al., 2020; Kweon & Yoon, 2024), bounding-boxes (Dai et al., 2015; Khoreva et al., 2017; Song et al., 2019), points (Bearman et al., 2016; Zhang et al., 2021; Wu et al., 2022; Liang et al., 2022; Wu et al., 2023), scribbles (Lin et al., 2016; Pan et al., 2021; Ke et al., 2021; Wu et al., 2022; Liang et al., 2022; Wu et al., 2023), and coarse annotations (Das et al., 2023b). Approaches based on image-level supervision primarily rely on class activation maps (CAMs) (Zhou et al., 2016) to localize objects. Several works (Ahn & Kwak, 2018; Sun et al., 2020) improve CAM quality using affinity-based propagation strategies, while more recent approaches (Kweon & Yoon, 2024) incorporate SAM to further refine CAM-based predictions. Although such methods perform well on relatively simple datasets such as PASCAL, generating reliable CAMs for more complex real-world scenes, such as Cityscapes (Cordts et al., 2016) and ADE20k (Zhou et al., 2017) remains challenging because of frequent multi-class co-occurrences. Bounding box supervision provides stronger spatial cues than image-level labels and therefore enables improved segmentation performance (Dai et al., 2015; Khoreva et al., 2017). In this work, we focus on *fine-grained* weak annotations, including points, scribbles, and coarse masks. Prior fine-grained WSSS methods (Pan et al., 2021; Ke et al., 2021; Wu et al., 2022; Liang et al., 2022; Wu et al., 2023; Zhang et al., 2021) are primarily evaluated on the relatively simpler PASCAL benchmark and typically consider only point or scribble supervision using conventional CNN-based architectures such as DeepLab. Recent fine-grained methods such as PBFA Chan et al., (2024), which augments scribble supervision with prototype-based feature hallucination, and SSL-Potts Zhang & Boykov (2025), which regularizes soft pseudo-labels with relaxed Potts objectives, remain scribble-centric, and PASCAL-focused. In contrast, we extend evaluation to more challenging benchmarks, including ADE20k (Zhou et al., 2017) and Cityscapes (Cordts et al., 2016), incorporate an additional coarse annotation setting, and adopt the state-of-the-art transformer-based segmentation architecture SegFormer (Xie et al., 2021). Furthermore, we leverage SAM as an instance prior to enhance the effectiveness of fine-grained weak annotations for weakly supervised semantic segmentation.

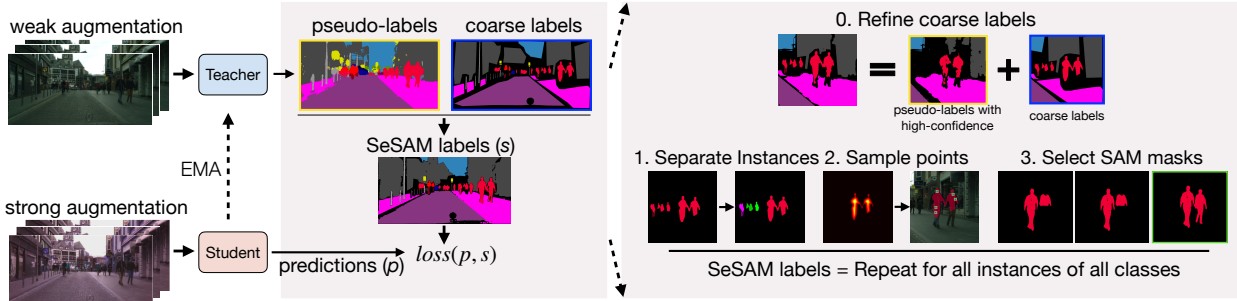

Figure 3: **SeSAM framework for coarse supervision**. First, pseudo-labels for training images are generated by the teacher model. Next, these pseudo-labels are then refined following the key steps 1-3, namely instance separation, point sampling and mask selection (see Section 3). To improve mask quality, we augment coarse labels with high-confidence pseudo-labels before applying SAM (step 0). The resulting refined labels, called SeSAM labels, are then used to train the student network.

**SAM as Segmentation Prior.** SAM (Kirillov et al., 2023; Ravi et al., 2024) is a segmentation foundation model that enables zero-shot segmentation using various prompts, including points, bounding-boxes, and masks. It has demonstrated strong performance across a wide range of fine-grained vision tasks, such as domain adaptation (Zhang et al., 2024; Chen et al., 2023b; Benigmim et al., 2024), object tracking (Yang et al., 2023; Li et al., 2024), referring segmentation (Yang et al., 2024; Sun et al., 2024; Zhang et al., 2023), and medical image segmentation (Ma et al., 2024; Shi et al., 2023; Deng et al., 2023; Hu et al., 2023). Despite its flexibility and strong zero-shot capabilities, directly applying SAM to downstream tasks remains challenging due to its dependence on user-provided prompts. Prior works (Yang et al., 2024; Xie et al., 2024; Chen et al., 2023a) address this limitation by using SAM's automatic mask generator, which produces masks for the entire image using a grid of point prompts. However, segmentation quality is highly sensitive to prompt placement, making grid-based prompting suboptimal. Alternative approaches guide prompt selection using network confidence (Kweon & Yoon, 2024) or feature similarity (Zhang et al., 2023). Nevertheless, selecting multiple prompts based solely on confidence or feature similarity does not consistently produce optimal segmentation results (see Fig. 2). Complementarily, ESC-Net Lee et al. (2025) tackles open-vocabulary segmentation by using SAM with text-derived prompts, whereas we use it with spatial weak annotations. In this work, we investigate SAM in the less-explored setting of weakly supervised semantic segmentation with fine-grained annotations, including points, scribbles, and coarse masks. We address key challenges in prompt selection, mask granularity, and semantic consistency, and integrate SAM within a semi-supervised framework to effectively leverage instance-level priors for semantic segmentation.

## 3 SeSAM Method

In this section, we begin with the preliminary work, describing the Segment Anything Model (SAM). We then introduce our SeSAM components (see Fig. 3 and Algorithm 1), addressing the challenges of leveraging instance priors from SAM for WSSS. First, we discuss how to use weak class labels to prompt SAM, followed by our sampling strategy for improved segmentation and then our approach to selecting the best mask granularity. Finally, we explain how we integrate semi-supervised learning with SAM-based instance priors during training.

### 3.1 SAM Preliminary

The Segment Anything Model (Kirillov et al., 2023) (SAM) is a foundational model for promptable segmentation, extensively utilized across various downstream applications. SAM consists of three primary modules: an image encoder, a prompt encoder, and a mask decoder. Given a prompt (e.g., points, boxes or masks), SAM generates a segmentation mask for a single object, part of an object, or group of connected objects. Specifically, the target image is first encoded into an image embedding, which is combined with the prompt

embedding and subsequently decoded into a segmentation mask. SAM produces three candidate masks with different levels of granularity, representing whole object, object part, and subpart, each accompanied by a confidence score. In our work, we retain the overall pipeline as is and instead focus particularly on the input point prompts and the three mask outputs. We determine how to effectively prompt the SAM model to enhance the quality of semantic segmentation using various weak annotations and how to train the model with ground-truth labels, SAM instance priors, and pseudo-labels simultaneously.

### 3.2 SAM as an Instance Prior

In this subsection, we outline the key components of our SeSAM framework. We first discuss the different types of weak supervision, followed by how we can leverage the instance priors of the SAM model with these weak-supervision labels.

**Types of Weak Supervision.** We perform WSSS using three types of fine-grained weak labels: point, scribble, and coarse annotations. Since point and scribble annotations are inherently sparse, we first convert them into coarser forms by leveraging SAM. Specifically, we first obtain an initial coarse mask by prompting SAM with the available weak labels. For point labels, the labeled point is passed directly to SAM as a prompt; for scribbles, we sample five points along the scribble using the same skeleton-based strategy (Section 3.2) and prompt SAM with them. To ensure high precision, we select the smallest-area SAM mask out of the three generated masks as our initial coarse mask. This coarse mask is then refined during training using our SeSAM pipeline.

**Classes vs. Instances.** While SAM can produce segmentation masks for given prompts, these masks cannot be directly used for semantic segmentation. Semantic segmentation requires labeling each pixel in an image as one of the predefined set of $N$ classes (e.g., one of 19 classes in Cityscapes), each representing all instances of the same class within the image. A naïve solution is to sample class points based on these class masks introduces ambiguity between different instances, as illustrated in Fig. 2 (e.g., class person). In this scenario, sampled points are distributed across all instances of a class, leading SAM to produce a single mask covering all sampled points. This often results in noisy masks and imprecise boundaries. To address this issue, we first identify the connected components (Tarjan, 1972) within each coarse class mask to separate individual instances. Let $M^{(c)} \in \{0,1\}^{H \times W}$ denote the coarse mask for class $c \in \{1, \ldots, N\}$, where $M^{(c)}(p) = 1$ indicates that pixel $p$ belongs to class $c$. Each class mask is decomposed into a set of $K_c$ disjoint instance masks using the connected component operator:

$$\{M_k^{(c)}\}_{k=1}^{K_c} = \text{CC}(M^{(c)}), \quad M^{(c)} = \bigcup_{k=1}^{K_c} M_k^{(c)}, \quad M_i^{(c)} \cap M_j^{(c)} = \varnothing \ \forall i \neq j. \tag{1}$$

We use the standard connected-component labeling from *scipy.ndimage.label* with a 4-connectivity structuring element to obtain the components. This operation is deterministic and hyperparameter-free. We then sample points for each identified instance, rather than across the entire class, and use these as prompts for SAM to generate instance-specific masks. This instance-based sampling strategy substantially improves mask accuracy and boundary precision.

**Point Sampling Matters.** To prompt SAM effectively, we rely on sampled points from each identified instance. However, for optimal refinement of the coarse labels associated with each instance, it is crucial to determine how we select $K$ points from within the mask to obtain the best possible segmentation proxies from SAM for subsequent training. As a naïve solution, random sampling, or sampling points solely from high-confidence regions, often produces masks that do not cover the full object extent. For example, as shown in Fig. 2 confidence-based sampling (Kweon & Yoon, 2024) for class 'person' results in segmenting only a part of the class instance. In order to ensure maximum coverage of the annotated region, we design a sampling strategy with three goals: (1) most of the points should be positioned near the central line of the mask to capture central details and connection between the different sub parts of the object; (2) some points should be close to the boundaries to refine edges; and (3) all $K$ sampled points should be uniformly distributed throughout the region. This approach ensures comprehensive coverage and enhances segmentation accuracy. To address the first criterion, we propose sampling points from the topological skeleton of the shape -

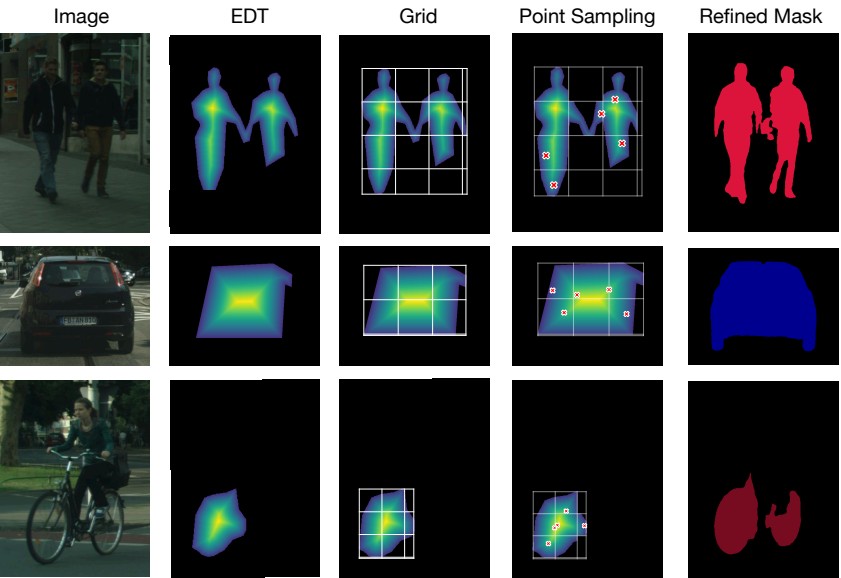

Figure 4: **Skeleton-based point sampling.** For each mask, we compute a normalized distance field (EDT) that peaks along the topological skeleton and decays toward boundaries. We tile the mask's bounding box into a grid (Eq. 2), sample K cells, and draw one point per cell from the distance field. Prompting SAM with these points yields the refined mask.

a simplified representation that captures the essential structure of a shape while preserving its topology. Formally, the skeleton of a 2D shape is a set of points lying at the "center" of the shape, equidistant from its boundaries. It effectively reduces the shape to a thin, connected line structure that retains important geometrical and topological properties, such as connectivity and general shape. For the second criterion, we define a probability distribution over the shape and its skeleton using a normalized distance field. This field is based on normalized Euclidean distances from each point within the shape to the nearest background point. With this setup, the skeleton forms the highest-probability region, while areas closer to the boundaries are less prominent. To satisfy the final criterion of uniformity, we employ a grid-based sampling approach. Specifically, we divide the bounding-box around the coarse annotations, with dimensions $S = s_1 \times s_2$ (where $s_1$ and $s_2$ are the side lengths), into a grid of size -

$$\lceil \frac{s_1}{\sqrt{\frac{S}{K}}} \rceil \times \lceil \frac{s_2}{\sqrt{\frac{S}{K}}} \rceil, \tag{2}$$

where $\lceil \cdot \rceil$ is the ceiling function. We then randomly select $K$ cells from this grid and sample one point from each cell based on the defined probability distribution. As shown in Fig. 4, the sampling strategy is globally uniform but locally skeleton-biased. The grid enforces coverage across the full object extent at the global scale, while within each cell the distance field concentrates probability on the skeleton - combining coverage with stability.

**Best SAM Mask.** To provide flexibility and minimize ambiguity, SAM generates three masks for each prompted object, covering whole, part, and subpart regions, each with an associated confidence score. A naïve solution would be to select the mask at random or with the highest confidence score predicted by SAM. However, such an approach often results in misleading coverage of the region (see Fig. 2, where selecting the mask with the highest SAM score results in part segmentation for the *'person'* class). Given the weakly supervised nature of our approach, selecting the most appropriate mask is essential to maximize both recall and precision. To balance these metrics, we propose a two-fold selection criterion: 1) we select the SAM mask that maximally covers the weak supervision area to maximize recall; 2) we exclude any excessively large SAM masks that significantly exceed the area of the initial weak supervision to maximize precision.

Table 1: Comparisons on PASCAL VOC 2012 using scribble and point labels with DeepLabv3+ and DeepLabV2 segmentation networks (ResNet-101 backbone). WvF (Ke et al., 2021) refers to weak vs fine supervision performance ratio. AGMM*: Our re-implementation. SAM-AMG[†] extends SAM-AMG Chen et al. (2023a) for point and scribble labels.

| | | Scribble | | Point | |
|---|---|---|---|---|---|
| Method | DeepLab | mIoU | WvF | mIoU | WvF |
| Pan et al. (Pan et al., 2021) | V2 | 74.6 | 96.0 | - | - |
| SPML (Ke et al., 2021) | V2 | 74.2 | 95.5 | - | - |
| A2GNN (Zhang et al., 2021) | V2 | 74.3 | 95.5 | 66.8 | 85.9 |
| Supervised | V3+ | 69.7 | 87.6 | 61.6 | 77.4 |
| Fixmatch (Sohn et al., 2020) | V3+ | 71.3 | 89.6 | 68.4 | 86.0 |
| DBFNet (Wu et al., 2022) | V3+ | 72.5 | 91.2 | 66.8 | 84.0 |
| TEL (Liang et al., 2022) | V3+ | 75.8 | 96.9 | 63.3 | 86.0 |
| AGMM (Wu et al., 2023) | V3+ | 76.4 | 96.1 | 69.6 | 87.5 |
| AGMM* (Wu et al., 2023) | V3+ | 76.7 | 96.4 | 68.7 | 86.4 |
| PBFA Chan et al. (2024) | V3+ | 76.2 | 95.8 | - | - |
| SSL-Potts Zhang & Boykov (2025) | V3+ | 78.1 | 98.2 | - | - |
| SAM-AMG[†] Chen et al. (2023a) | V3+ | 76.2 | 95.8 | 67.2 | 84.5 |
| SAM baseline | V3+ | 38.4 | 48.3 | 71.1 | 89.4 |
| Ours | V3+ | 78.1±0.2 | 98.2 | 75.2±0.1 | 94.6 |

For each prompt, SAM produces three candidate masks $\{S_i\}_{i=1}^3$. Given a weak mask $W$, we compute a coverage score ($r_i$) and compatibility score ($p_i$) given by -

$$r_i = \frac{|S_i \cap W|}{|W|} \quad p_i = \frac{|S_i \cap W|}{|S_i|} \tag{3}$$

We retain candidate masks that satisfy $r_i \geq \tau_1$ and $p_i \geq \tau_2$, and select the valid mask with the highest compatibility score. In our experiments, we set $(\tau_1, \tau_2) = (0.3, 0.7)$. If no candidate satisfies both thresholds, we select the mask with the highest compatibility score. This criterion avoids selecting overly partial masks while preserving consistency with the weak annotation.

### 3.3 SAM & Semi-Supervised Learning

In our approach, we leverage three types of labels: ground-truth weak annotations, SAM-generated labels, and pseudo-labels. For robust pseudo-labels, our method employs weak-strong augmentations to enhance semi-supervised learning, similarly to Fixmatch (Sohn et al., 2020). This framework leverages SAM to dynamically generate segmentation masks, supporting flexibility in training while improving segmentation quality, particularly where ground-truth labels are coarse or incomplete. The ground-truth consists of weak annotations in the form of coarse semantic segmentation masks, which often lack precise boundaries and may omit small objects due to labeling constraints. During training, SAM is applied only to weakly augmented images, adding variations and scalability to SAM-derived masks. This helps in improving segmentation accuracy by incorporating diverse views of the segmentation boundaries. Following (Sohn et al., 2020), we utilize high-confidence pseudo-labels filtered using threshold $\Theta_1$. These pseudo-labels effectively address segmentation for small objects that lack ground-truth labels and enhance object coverage over the training process.

**Refined Mask Sampling.** Since the coarse mask often fails to cover the entire class instance, sampling points exclusively from these regions can be suboptimal. To enable sampling from the full object extent, we extend coarse masks using pseudo-labels filtered with a stricter threshold, $\Theta_2 > \Theta_1$. Sampling from this pseudo-labeled extension improves recovery of the complete instance mask. To further improve efficiency, we dynamically resample SAM masks every $M = 4$ iterations to reflect updates in the coarse annotations. This periodic resampling produces our SeSAM masks (Fig. 3), allowing the masks to adapt to progressively refined and expanded annotations, ensuring consistency across evolving masks. We provide ablations on the sampling-frequency trade-off in Appendix Table 7 and runtime comparisons in Appendix Table 6.

Table 2: Comparison on Cityscapes (Cordts et al., 2016), ADE20k (Zhou et al., 2017), and PASCAL VOC 2012 (Everingham et al., 2010) with SegFormer (Xie et al., 2021) and DeepLabV3+. AGMM* denotes our reimplementation of AGMM (Wu et al., 2023). SAM-AMG† extends SAM-AMG Chen et al. (2023a) for point, scribble and coarse labels. The SAM baseline naïvely uses weak labels as prompts to SAM (see Section 4.1). SegFormer uses MiT-B0, whereas DeepLabV3+ uses ResNet-101 (He et al., 2016).

| Annotation | Method | SegFormer | | | DeepLabV3+ | |
|---|---|---|---|---|---|---|
| | | Cityscapes | ADE20k | PASCAL | Cityscapes | ADE20k |
| **Point** | Supervised | 51.8 | 26.5 | 61.5 | 52.2 | 29.5 |
| | Fixmatch (Sohn et al., 2020) | 53.1 | 27.2 | 63.6 | 52.9 | 29.8 |
| | AGMM* (Wu et al., 2023) | 53.4 | - | 56.6 | 52.4 | - |
| | SAM-AMG† Chen et al. (2023a) | 47.0 | 26.6 | 60.0 | 53.8 | 30.3 |
| | SAM baseline | 52.1 | 30.9 | 60.9 | 53.9 | 34.9 |
| | Ours | **58.5**±0.3 | **32.5**±0.1 | **67.4**±0.3 | **61.8**±0.1 | **36.3**±0.1 |
| **Scribble** | Supervised | 62.6 | 35.5 | 66.4 | 63.8 | 39.6 |
| | Fixmatch (Sohn et al., 2020) | 64.2 | 35.8 | 67.1 | 64.8 | 40.4 |
| | AGMM* (Wu et al., 2023) | 56.7 | - | 68.1 | 67.4 | - |
| | PBFA Chan et al. (2024) | - | - | - | 67.5 | 40.0 |
| | SSL-Potts Zhang & Boykov (2025) | - | - | - | 72.4 | 39.7 |
| | SAM-AMG† Chen et al. (2023a) | 60.9 | 32.3 | 67.7 | 70.2 | 40.6 |
| | SAM baseline | 27.0 | 11.1 | 35.5 | 28.1 | 11.7 |
| | Ours | **71.6**±0.2 | **37.0**±0.1 | **71.5**±0.3 | **75.3**±0.1 | **43.3**±0.2 |
| **Coarse** | Supervised | 65.2 | 35.7 | 62.8 | 67.9 | 40.9 |
| | Fixmatch (Sohn et al., 2020) | 66.7 | 36.7 | 63.6 | 68.9 | 41.3 |
| | AGMM* (Wu et al., 2023) | 57.1 | - | 58.6 | 70.6 | - |
| | SAM-AMG† Chen et al. (2023a) | 61.9 | 32.1 | 66.5 | 69.9 | 39.5 |
| | SAM baseline | 43.1 | 24.2 | 41.4 | 48.5 | 27.0 |
| | Ours | **69.9**±0.1 | **37.3**±0.1 | **70.2**±0.2 | **72.5**±0.3 | **43.2**±0.1 |
| Fine | Supervised | 76.0 | 38.1 | 73.5 | 80.2 | 44.6 |

**Overall Loss Function.** We train the segmentation network using three sources of supervision: weak labels, SAM-derived labels, and teacher pseudo-labels. The overall objective is

$$\mathcal{L} = \mathcal{L}_{\mathrm{coarse}} + \lambda_1 \mathcal{L}_{\mathrm{SAM}} + \lambda_2 \mathcal{L}_{\mathrm{pseudo}} \qquad (4)$$

where $\lambda_1$ and $\lambda_2$ are weights for loss components with SAM-derived labels and teacher pseudo-labels respectively. Each loss is computed only on the pixels labeled by its corresponding supervision source, while unlabeled pixels are ignored. When multiple supervision sources overlap, weak labels take precedence over SAM-derived labels, and SAM-derived labels take precedence over pseudo-labels.

# 4 Experiments

In this section, we discuss the extensive experiments to show the effectiveness of our SeSAM framework.

**Datasets and metrics used.** Following prior works (Pan et al., 2021; Ke et al., 2021; Zhang et al., 2021; Wu et al., 2022; Liang et al., 2022; Wu et al., 2023), we evaluate our method on PASCAL VOC 2012 (Everingham et al., 2010) and further extend evaluation to the challenging ADE20k (Zhou et al., 2017) and Cityscapes (Cordts et al., 2016). PASCAL VOC 2012 contains 10,582 training images and 1,449 validation images, with 20 annotated classes. ADE20k features both indoor and outdoor scenes, with 150 labeled classes and approximately 20,000 training and 5,000 validation images. Cityscapes comprises urban scenes with 2,975 training images and 500 validation images across 19 annotated classes. For all the datasets, we obtain the point annotations by randomly sampling one point per class per image following (Paul et al., 2020). Scribble annotations for these datasets are provided by (Boettcher et al., 2024; Lin et al., 2016). While Cityscapes provides manually annotated coarse labels, we obtain the coarse annotation for ADE20k and PASCAL VOC 2012 dataset following (Das et al., 2023b). For evaluation, we use the mean Intersection over Union (mIoU) metric and Weak over Fine supervision performance ratio (WvF).

**Implementation details.** We present our results on two segmentation networks: SegFormer (Xie et al., 2021) with MiT-B0 backbone and DeepLabv3+ (Chen et al., 2018) with ResNet-101 backbone. We use standard crop sizes (Xie et al., 2021) of 1024×1024 for Cityscapes and 512×512 for ADE20k and PASCAL VOC 2012 during training. For the Segment Anything model (SAM) (Kirillov et al., 2023), we employ the SAM ViT-B variant in our experiments, keeping the model frozen throughout training. We train for 120k iterations with a batch size of 8 for Cityscapes, 160k iterations with batch size of 16 for ADE20k, and 60k iterations with batch size of 16 for PASCAL VOC 2012 dataset. We use the AdamW optimizer with an initial learning rate of $1 \times 10^{-4}$. To generate robust pseudo-labels, we adopt a standard teacher-student framework (Sohn et al., 2020), where the teacher model is an exponential moving average of the student model (EMA decay rate, $\alpha = 0.999$). We sample $K = 5$ points from each instance mask. Following Fixmatch (Sohn et al., 2020) we choose $\Theta_1 = 0.96$. For loss terms weighting, we select $\lambda_1 = 1.0$ and $\lambda_2 = 0.01$. For confident pseudo-labels used for coarse extension, we use higher threshold for $\Theta_2$ as 0.98.

**Annotation costs.** For the Cityscapes dataset, annotating with fine labels takes around 90 min per image (Cordts et al., 2016) , while coarse annotation requires around 7 min (Cordts et al., 2016). Point annotations are significantly faster at ∼45 seconds (Paul et al., 2020; Das et al., 2023a) per image. Following (Lin et al., 2016), the annotation cost of scribble labels is estimated to be around 2.2 min per image.

## 4.1 Comparison with baselines

Prior works (Pan et al., 2021; Ke et al., 2021; Zhang et al., 2021; Wu et al., 2022; Liang et al., 2022; Wu et al., 2023) have primarily evaluated WSSS on the relatively simple PASCAL VOC 2012 dataset using only two weak labels (point and coarse) and CNN-based segmentation networks (DeepLabv2/DeepLabv3+). In contrast, we extend the evaluation to more challenging Cityscapes and ADE20k datasets, using three weak labels (including an additional coarse label) and employing the transformer-based SegFormer network. In Table 1 we first compare SeSAM's performance on standard PASCAL VOC 2012 benchmark. Next, in Table 2 we provide comparison on challenging Cityscapes and ADE20k datasets. For this comparison, we take the best performing prior work, AGMM (Wu et al., 2023) and reimplement it on this setting.

In Table 1, we present comparisons to the previous work on a simple object-oriented PASCAL dataset. Our method significantly outperforms all prior work (Pan et al., 2021; Ke et al., 2021; Zhang et al., 2021; Wu et al., 2022; Liang et al., 2022; Wu et al., 2023) with both scribble and point labeling. In particular, SeSAM shows better performance than the best prior work AGMM by 1.7% and 5.7% mIoU with scribble and point labels, respectively, achieving best weak vs fine (WvF) supervision performance ratio. Further, we go beyond simple object-centric PASCAL dataset and evaluate on more realistic and challenging scenarios, such as in Cityscapes and ADE20k datasets. In Table 2, we evaluate our proposed method SeSAM on these datasets with CNN-based architecture (DeepLabV3+) and transformer-based architecture (SegFormer) and show that SeSAM outperforms the baselines and the best previous method AGMM by a substantial margin. Note that AGMM re-engineers the DeepLabv3+ network and their method does not generalize well to transformer based segmentation networks. It further becomes computationally infeasible with increasing number of classes. For ADE20k with 150 classes it requires around 259.2 GB GPU memory for DeepLabV3+(ResNet-101) compared to 75.5 GB for PASCAL dataset (21 classes).

While it is not straightforward to apply SAM using weak labels for the WSSS task (see Fig. 2), we constructed a naïve baseline that directly uses the weak labels (point (Bearman et al., 2016), scribble (Lin et al., 2016), and coarse (Das et al., 2023b)) to prompt SAM and generate labels for training the segmentation model. In particular, for point-based labels, each labeled point is provided as input to SAM. Scribbles are treated as a collection of points and used as such for prompting SAM, while coarse labels are provided as mask prompts. We pre-generate labels using SAM with these weak label prompts and train the segmentation model with semi-supervised pipeline. This isolates the contribution of our three-step refinement from the semi-supervised training itself. We observe a major drop in performance particularly for scribble and coarse labels (e.g., a drop of 44.6% mIoU for Cityscapes with scribbles), suggesting naïvely using SAM generates noisy labels. Additionally, we compare with another baseline (SAM-AMG), where we extend Chen et al. (2023a) for point, scribble and coarse labels and use SAM's automatic mask generator to generate SAM masks. In particular, dense grid prompts produce class-agnostic masks over the whole image, and each mask

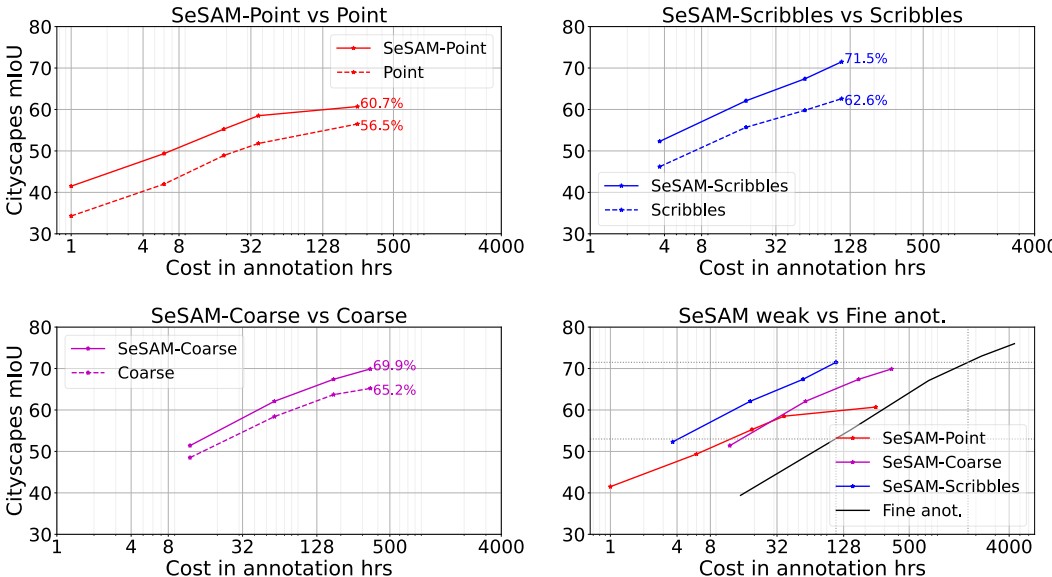

Figure 5: **Annotation cost vs. performance plot**. We compare the performance at different budgets for each of the weak labels (point, scribble and coarse), and compare the cost-effectiveness of our method with fine annotation (bottom-left). In particular, scribble supervision reaches performance *comparable* to fine-label supervision using only 6% of the annotation budget. Whereas, in a low-budget setting (109 hrs), it outperforms fine annotations by 18.5%.

is assigned the class of the weak label with which it has maximal overlap. The network is then trained with such labels. SAM-AMG performs poorly compared to our approach showing importance of our our point sampling approach. Further, when compared to the upper bound performance, i.e. supervised baseline on fine labels, we observe that our approach successfully narrows the performance gap between weak and fine annotations. For example, our coarse setting on ADE20k dataset achieves 37.3% mIoU whereas supervised training on full fine annotation yields 38.1%. Overall, our proposed framework outperforms the prior works and baselines for all of the weak labels (point, coarse, and scribbles) showing the effectiveness of incorporating SAM priors in the WSSS setting.

**Qualitative Comparison.** We provide the qualitative comparison of SeSAM and supervised learning with different weak labels (point, scribble, and coarse) on Cityscapes dataset using SegFormer segmentation network in Fig. 6. We observe that with additional SAM instance priors, SeSAM has better prediction along boundaries (e.g. car). It also correctly predicts the class road for scribble and point weak labels.

### 4.2 Cost vs performance trade-off

In this experiment we show the cost-effectiveness of our approach. We perform this comparison on Cityscapes dataset in Fig. 5. We sample 100, 500, 1500, 2975 images from the Cityscapes fine and coarse training sets. The cost of annotations are computed following the annotation costs mentioned in Section 4. For example, annotating 100 images with fine labels costs 150hrs, coarse labels cost 11.6 hrs, scribble labels 3.6 hrs, whereas point label costs 1.2 hrs.

We make two comparisons. First, we compare the performances of different weak labels in different labeling budgets. Then, we compare the performance with supervised learning on fine labels and different weak labels. For the first comparison, we observe that scribble labels outperform the coarse and point labels. For the Cityscapes dataset, we generated the initial coarse masks from scribbles as discussed in Section 3. The better performance can be justified by the higher quality of "SAM generated" coarse masks than of the

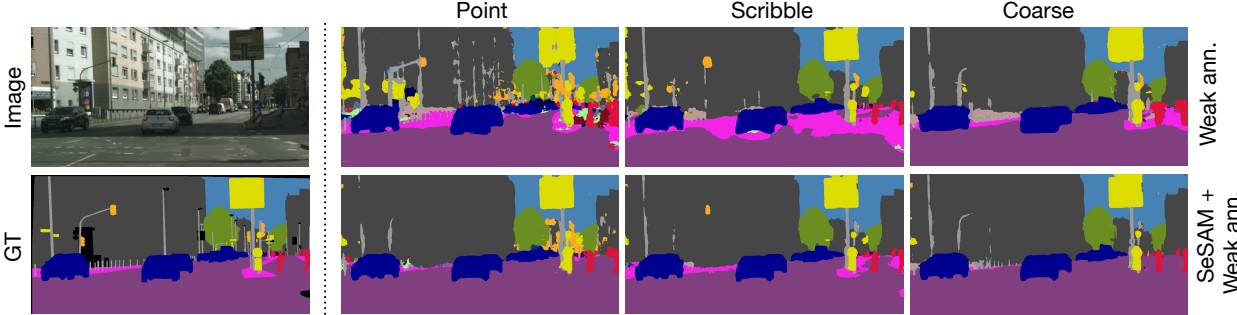

Figure 6: **Qualitative performance comparison**. We conduct a qualitative performance comparison of our framework with supervised models using various weak labels, including point, scribble, and coarse annotations. Overall, our method demonstrates superior class segmentation, particularly with enhanced prediction along boundaries (see car).

manually labeled coarse annotations (see *Appendix*). For the second comparison, we observe that scribble labeling is a better alternative to fine annotations in limited budget setting. In particular, scribble annotation achieves similar performance to fine annotations using only 6% of the annotation budget (109 hrs vs 4463 hrs). In limited budget setting (e.g. 109 hrs), scribble outperforms fine annotation by 18.5%. A similar conclusion can be drawn for other weak labels. Overall, weak labels are cost-effective alternatives to fine labels, particularly in constrained budgets.

Table 3: **Ablation studies of SeSAM.** We study the contribution of the proposed components, instance mask separation, and refined-mask-based point sampling. Base refers to the FixMatch (Sohn et al., 2020)-based segmentation network. Experiments on Cityscapes with SegFormer-B0.

| Base | Instance | Mask Selection | Prompt Sampling | Refined Sampling | Coarse |
|------|----------|----------------|-----------------|------------------|--------|
| ✓ | | | | | 65.2 |
| ✓ | ✓ | | | | 68.6 |
| ✓ | ✓ | ✓ | | | 68.9 |
| ✓ | ✓ | | ✓ | | 69.1 |
| ✓ | ✓ | ✓ | ✓ | | 69.4 |
| ✓ | ✓ | ✓ | ✓ | ✓ | **69.9** |

(a) Ablation of SeSAM components.

| | Coarse |
|---|---|
| W/o instance | 67.9 |
| With instance | **69.9** |

(b) Effect of instance mask separation.

| | Coarse | Point |
|---|---|---|
| W/o refined | 69.4 | 56.0 |
| With refined | **69.9** | **58.5** |

(c) Effect of refined-mask-based point sampling.

## 4.3 Ablation Studies and Analysis

In this subsection, we analyze the key components of SeSAM through a series of ablation studies. We first ablate the core components of the framework, and then examine the main design choices introduced to address the challenges of using SAM for weakly supervised semantic segmentation, including instance separation, mask selection, point sampling, and prompt type. Finally, we study the impact of sampling point prompts from refined masks during training.

**SeSAM components.** In this experiment (see Table 3, left), we ablate the individual components of SeSAM. Starting from FixMatch (Sohn et al., 2020) as the baseline, we progressively incorporate SAM-based instance priors. Instance separation alone provides a significant gain of 3.4% over FixMatch. For this variant, we use naïve point sampling and mask selection strategies; specifically, we sample high-confidence point prompts (top-$k$) and select the highest-scoring masks. We further observe that our proposed mask selection strategy and uniform probabilistic prompt sampling each improve performance individually, while their combination yields an additional gain of 0.8%. Finally, refined mask sampling provides a further improvement of 0.5%. Overall, the ablation confirms that the proposed components are complementary and jointly yield the best performance.

Table 4: **Analysis of SeSAM design choices.** We study the effect of different point sampling strategies, mask selection strategies, and prompt types. Experiments on Cityscapes with SegFormer-B0.

| Method | Coarse |
|---|---|
| High confidence | 68.6 |
| Center | 68.4 |
| Boundary | 68.8 |
| Random | 69.4 |
| Ours | **69.9** |

(a) Sampling strategies.

| Method | Coarse |
|---|---|
| Best score | 69.0 |
| Random | 68.8 |
| Ours | **69.9** |

(b) Effect of mask selection strategy.

| Label | mIoU |
|---|---|
| BBox | 68.3 |
| Mask | 63.4 |
| Point | **69.9** |

(c) Effect of prompt type.

**Instance Separation.** Qualitatively, we observe that separating instances within class masks improves segmentation quality with SAM, as shown in Fig. 2. Quantitatively, performance drops by 2% relative to a baseline without instance separation (see Table 3, middle), highlighting the importance of this design choice. This suggests that sampling points from instance-level masks, rather than class masks, provides better prompts and leads to more accurate mask refinement.

**Selection of SAM mask granularity.** The naïve baseline is to select the candidate mask with the highest SAM score. However, as shown in Fig. 2, the highest-scoring mask often corresponds to a partial object rather than the desired full instance. Selecting the mask based on the best score results in a reduction in performance by 0.9 mIoU, while random mask selection reduces performance by 1.2 mIoU. Overall, our weak-label-aware mask selection strategy consistently outperforms these baselines (see Table 4).

**Point sampling strategy.** In this experiment, we compare various ways of sampling points from coarse masks for better SAM segmentation. As shown in Fig. 2, sampling of points based on network confidence is suboptimal. We further quantify this ablation in Table 4 left. We additionally compare with center sampling, boundary sampling, and random sampling. Center sampling selects pixels near the distance-transform maxima; boundary sampling selects pixels close to object boundaries; and random sampling draws uniformly from the foreground. Among these strategies, our skeleton-aware probabilistic sampling achieves the best performance.

**Sampling from refined mask.** Our framework iteratively refines SAM masks by extending the coarse mask with pseudo-labels as discussed in Section 3. We further allow sampling of point prompts from these pseudo-label extension of coarse masks. We ablate (see Table 3 right) this addition with a baseline that does not allow sampling from pseudo-labels. We observe that allowing sampling of point prompts from pseudo-label extensions can further boost the performance. In particular, we observe a bigger performance gain for point label (2.5%) than coarse label (0.5%), as point labels allow a larger extension with pseudo-labels than coarse labels.

**Why do we use points for prompting?** In this experiment, we compare different choices of prompts used by SeSAM. For point prompts, we sample points from the coarse mask based on our uniform skeleton-based probabilistic sampling discussed in Section 3. For bounding-boxes, we compute the bounding-box around the coarse mask and use it as a prompt for SAM. For mask prompt, we directly use the coarse mask as prompt. Mask prompts fail to generate meaningful SAM masks, whereas bounding-box prompt restricts the mask to the box (see *Appendix* Fig. 9). Overall, point prompting performs better (see Table 4) than other prompting types.

**Comparison across different SAM versions.** We conduct an additional experiment to compare the mask generation quality of different SAM versions: SAM (Kirillov et al., 2023), SAM 2 (Ravi et al., 2024), and SAM 3 (Carion et al., 2025). We use the Cityscapes validation set and apply the sampling strategy from Section 3 to sample points from coarse `instance` labels. These points are used to prompt SAM, which refines the coarse masks by expanding them with SAM-generated masks. We evaluate refinement quality using precision and recall computed on labels added by SAM. As shown in Table 5, the original SAM achieves the best trade-off between precision and recall for image segmentation, whereas newer versions (SAM 2 and SAM 3), despite their additional capabilities such as video and concept-based segmentation, perform slightly

Table 5: **Comparison across SAM versions.** We compare the mask generation quality of different SAM versions using precision and recall. Although newer versions (SAM 2 and SAM 3) introduce additional capabilities beyond image segmentation (Ravi et al., 2024; Carion et al., 2025), the original SAM provides the best trade-off between precision and recall for image-based segmentation. Experiments on Cityscapes dataset.

|  | SAM (Kirillov et al., 2023) | SAM 2 (Ravi et al., 2024) | SAM 3 (Carion et al., 2025) |
|---|---|---|---|
| Precision | 92.2 | **92.8** | 92.1 |
| Recall | **62.7** | 61.8 | 61.9 |
| F1 | **74.7** | 74.2 | 73.9 |

worse in this setting. For a fair comparison, we use: SAM ViT-B for SAM, SAM 2.1 Hiera-B+ for SAM 2, and the only publicly released model for SAM 3.

Table 6: **Runtime and memory comparison** on Cityscapes using DeepLabv3+. We report training time per iteration, GPU memory usage, and mIoU.

|  | Training time (sec/iter) | GPU memory (Gb) | Performance (mIoU) |
|---|---|---|---|
| Fixmatch (Sohn et al., 2020) | 0.24 | 22.3 | 68.9 |
| AGMM (Wu et al., 2023) | 1.56 | 60.8 | 70.6 |
| Ours | 0.81 | 22.7 | 72.4 |

**Efficiency Analysis.** We evaluate SeSAM in terms of training time and GPU memory usage, and compare it with the prior WSSS method AGMM (Wu et al., 2023) and the semi-supervised semantic segmentation baseline FixMatch (Sohn et al., 2020). All experiments are conducted on the Cityscapes dataset with a batch size of 4 and a crop size of $1024 \times 1024$. For a fair comparison with the prior state of the art, AGMM, we adopt the same DeepLabV3+ backbone. As shown in Table 6, SeSAM is substantially more efficient than AGMM, reducing training time from 1.56 to 0.81 sec/iter and GPU memory usage from 60.8 to 22.7 GB, while also improving mIoU from 70.6 to 72.4. Relative to the lighter FixMatch baseline, SeSAM requires longer training time due to SAM-based mask generation, but preserves similar memory usage (22.7 vs. 22.3 GB) and yields a notable gain in mIoU (72.4 vs. 68.9 mIoU). Overall, these results show that SeSAM substantially improves over the prior WSSS method AGMM and achieves higher accuracy than FixMatch with similar memory usage, at the cost of increased training time.

## 5 Conclusion

This work presents SeSAM, a weakly supervised semantic segmentation framework that leverages instance priors from the Segment Anything Model (SAM). We address key challenges in adapting SAM for weak supervision. First, we show how to prompt SAM using weak class labels, proposing an approach that separates class masks into individual instances. Next, we show that uniform point sampling along the mask skeleton yields optimal mask quality. To address ambiguity in selecting the desired mask granularity, we introduce a metric based on intersection with weak labels. Finally, we integrate diverse label types—ground-truth weak labels, SAM-generated labels, and pseudo-labels within a semi-supervised framework, enhancing mask quality by sampling points from pseudo-label extensions of the weak labels. Overall, our framework operates with various weak labels: points, scribbles, and coarse annotations, narrowing the performance gap with fully annotated data at a fraction of the annotation cost.

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
