# OpenReview forum: "Do Instance Priors Help Weakly Supervised Semantic Segmentation?"
_TMLR — Under review for TMLR_

### Review · Reviewer_Mx9m · 2026-05-11

**Summary Of Contributions:**

The paper proposes SeSAM, a weakly-supervised semantic segmentation framework based on SAM, with a focus on weak supervision signals including points, scribbles, and coarse annotations.

The claimed contributions include:

(1) a three-step pipeline to adapt SAM for weakly-supervised semantic segmentation;

(2) a semi-supervised learning framework;

(3) analysis of cost-performance trade-off across weak supervision signals, including point, scribble, and coarse labels;

(3) superior performance and generalization across weak supervision signals.

**Audience:**

Yes

**Audience Explanation:**

The paper studies weakly-supervised semantic segmentation task, which is of interest to those working on either (weakly supervised learning/semantic segmentation) field.

**Claims And Evidence:**

Yes

**Claims Explanation:**

The claims are partially supported by the experiments. In particular, the claim of outperforming previous baselines is not supported convincingly by Table 1, as the comparisons are restricted to works published in/before 2023. The paper needs to strengthen this claim by adding more advanced baselines.

**Requested Changes:**

1. The paper could benefit from strengthening its motivation for adapting the Instance-based SAM model for Semantic Segmentation (SS) against Instance Segmentation (IS). Given the Instance priors, would it be natural (more directly) to adapt the SAM for weakly supervised IS instead of SS?


2. Section 3.2, (paragraph Point Sampling Matters). The goals of the sampling strategy require clarification: The first goal requires most points to be near the central line of the masks, while the third goal requires a uniform distribution over the whole region. This sounds contradictory.

-- While Fig.12 is helpful, I would suggest adding visualizations of individual implementations of these three goals for a comparison.

3. The paper needs to add more advanced baselines for comparisons in order to strengthen its performance claim.


4. The first two contributions (i.e., three-stage pipeline and the semi-supervised framework) seem to overlap. My understanding is that the framework is a weakly-supervised approach, but I do not understand how semi-supervised learning is implemented (e.g., from the overall loss function of Section 3.3).

5. The paper may benefit from discussing its possible failure cases and limitations, and the discussion of generalization, in particular, its multiple hyperparameters are not discussed.

6. SAM does not work well with sparse inputs. While the paper addresses it, the question is whether a stronger baseline that is built upon SAM's mask generator/grid prompts works. The paper may need to strengthen its first contribution claim by adding straightforward yet more powerful baselines for comparisons.

7. Some related works can be added for discussion/comparisons, e.g.,

[A] Segment Anything Model (SAM) Enhanced Pseudo Labels for Weakly Supervised Semantic Segmentation, arXiv 2023

[B] Effective SAM Combination for Open-Vocabulary Semantic Segmentation, CVPR 2025

---

> ### Author Response · Authors · 2026-07-05
>
> We thank Reviewer Mx9m for the careful review, and are glad that the reviewer found the problem of broad interest and acknowledged our gains. We strengthen the comparisons with two recent baselines (ICML'24, CVPR'25) and address each point below.
>
> ---
>
> **1. Why adapt instance-based SAM to semantic segmentation rather than instance segmentation?**
>
> Thank you for this question. We argue that semantic segmentation and not instance segmentation is the natural target for the following reasons:
>
> (a) **The supervision itself is class-level, not instance-level.** Our weak labels (points, scribbles, and coarse masks) carry only class labels. Scribbles and coarse labels span multiple instance objects, and provide no instance correspondence.
>
> (b) **The task objectives differ fundamentally.** Semantic segmentation requires a dense, class-consistent labeling of every pixel, including "stuff" classes (road, sky, vegetation, terrain) for which instance segmentation is ill-defined. SAM, in contrast, outputs independent, class-agnostic masks of varying granularity (whole/part/subpart). These instance masks must still be (i) assigned to semantic classes, (ii) merged across instances of the same class, and (iii) reconciled where masks of different granularity conflict. Our experiments show this gap is non-trivial: the naïve SAM baseline drops by up to 44.6 mIoU on Cityscapes with scribbles (Tab. 2).
>
> (c) **This gap is precisely our contribution.** SeSAM converts SAM's instance priors into semantically consistent supervision (instance separation → skeleton-based point sampling → weak-label-aware mask selection), and integrates them into training. So rather than re-purposing SAM for the task it is already closest to (instance segmentation), we study whether its instance priors transfer to the harder, complementary setting of dense semantic labeling with weak class-level supervision and show that they do, but only with careful adaptation.
>
> ---
>
> **2. Sampling strategy clarification**
>
> We agree the current wording can be a bit confusing, and we have revised Sec. 3.2 for clarity. There is no contradiction: the sampling is globally uniform but locally skeleton-biased, i.e., the two goals operate at different spatial scales:
>
> - **Global scale:** we divide the instance bounding box into a grid (Eq. 2) and draw one point per selected cell. This guarantees `coverage`, i.e. the K points are spread across the full object extent, so SAM is anchored to all parts of the object.
> - **Local scale:** `within` each cell, points are drawn from the normalized distance-transform distribution, which concentrates probability on the topological skeleton while still leaving non-zero mass near boundaries.
>
> **Why both are needed:** center/skeleton-only sampling lacks boundary cues and yields conservative masks that under-cover the object; boundary-only sampling is unstable, since near-edge points can cause SAM to over-segment. Our strategy combines coverage (grid) and stability (skeleton bias). This is supported quantitatively by Table 4a: center-only (68.4), boundary-only (68.8), underperform wrt our strategy (69.9 mIoU).
>
> **Visualizations:** Following the reviewer's helpful suggestion, we have extended Fig. 4 (in main paper) with side-by-side visualizations of the individual implementations for better understanding.
>
> ---

---

> ### Author Response · Authors · 2026-07-05
>
> **3. More advanced baselines to strengthen the performance claim.**
>
> We thank the reviewer for the question. We would first like to clarify the state of the field, and then report new comparisons.
>
> **State of the field.** Our work targets WSSS with fine-grained weak labels (points, scribbles, coarse masks). The large majority of recent WSSS literature: including most 2024-2025 SAM-based works, operates with image-level labels and is evaluated only on PASCAL (see the comprehensive list at - https://github.com/pengtaojiang/awesome-weakly-supervised-semantic-segmentation-papers
>
> As we show in the supplement (Sec. A.2, Fig. 11), image-level methods do not transfer to real-world scenes: the strong image-label SAM-based method, SAM2CAM (CVPR'24), achieves only 5.8 mIoU on Cityscapes vs. 61.8 for SeSAM with point labels under the same DeepLabv3+ network, because frequent class co-occurrence (e.g., road+car in nearly every image) breaks CAM localization. This is why our comparisons focus on fine-grained-label, with AGMM as a strong baseline at submission time.
>
> **New comparisons.** Following an exhaustive search, we identified two recent fine-grained WSSS works and have added them as baselines:
>
> 1. PBFA - Scribble-Supervised Semantic Segmentation with Prototype-based Feature Augmentation (ICML'24). Augments scribble supervision by hallucinating features around class prototypes mined from the labeled scribble pixels. Like most prior fine-grained work, it is designed and evaluated for scribbles on PASCAL only, and uses no foundation-model priors. We compare on PASCAL and additionally re-implement and extend it to Cityscapes and ADE20k.
>
> 2. SSL-Potts - Soft Self-labeling and Potts Relaxations for Weakly-Supervised Segmentation (CVPR'25). Improves scribble-supervised training via soft pseudo-labels regularized with relaxed Potts/CRF objectives. It is likewise scribble-centric and prior-free. We compare directly on all three datasets.
>
> Neither prior method handles point or coarse annotations or the challenging Cityscapes/ADE20k benchmarks out of the box.
>
> | Method | PASCAL (Scribble) | Cityscapes (Scribble) | ADE20k (Scribble) |
> |---|---|---|---|
> | PBFA (ICML'24) | 76.2 | 67.5 | 40.0 |
> | SSL-Potts (CVPR'25) | 78.1 | 72.4 | 39.7 |
> | SeSAM (Ours) | 78.1 | 75.3 | 43.3 |
>
> Overall, SeSAM outperforms both recent baselines, and we will add these comparisons to Tables 1-2 and the Related Work discussion in the revision.
>
> ---
>
> **4. Overlap between the first two contributions; how is semi-supervised learning implemented?**
>
> We apologize for the confusion. The two contributions operate at different levels -
>
> **Contribution 1:** The three-step SAM adaptation pipeline (label generation): instance separation, skeleton-based point sampling, and weak-label-aware mask selection (Sec. 3.2). Its output is a set of refined, semantically consistent SeSAM labels produced from the weak annotations. This pipeline is training-framework-agnostic.
>
> **Contribution 2:** The semi-supervised training framework (label consumption): how those labels are used during training (Sec. 3.3, Fig. 3). We adopt a semi-supervised teacher-student training setup (FixMatch): the teacher (EMA of the student) produces pseudo-labels on weakly augmented images; the student is trained on strongly augmented views. The total loss is L = L_coarse + λ₁·L_SAM + λ₂·L_pseudo, where each term is computed only on pixels labeled by its supervision source, with the precedence weak > SAM > pseudo on overlapping pixels.
>
> **"Semi-Supervised in a Weakly-Supervised Method":** The SSL operates at the pixel level, we extend FixMatch (a semi-supervised learning method) for learning at pixel level. Under weak supervision, every image is only partially labeled - labeled pixels (weak labels + SeSAM labels) play the role of the "labeled set" while the remaining unlabeled pixels are supervised by high-confidence teacher pseudo-labels, exactly the consistency-and-confidence mechanism of FixMatch.
>
> ---

---

> ### Author Response · Authors · 2026-07-05
>
> **5. Failure cases, limitations, hyperparameter generalization**
>
> **Hyperparameter generalization.** This is partially covered in the supplement, Appendix Sec. A.2 ("Robustness of hyperparameters") ablates the number of sampled points K and the mask-selection thresholds (τ₁, τ₂) across both Cityscapes and ADE20k, showing that the same setting (K = 5, τ₁ = 0.3, τ₂ = 0.7) is optimal on both datasets (Figs. 7–8). The remaining hyperparameters (Θ, λ, EMA decay) are inherited from FixMatch. The SAM resampling period M is ablated in Table 7 with optimum at M = 4. The PASCAL and ADE20k experiments reuse the Cityscapes settings without re-tuning.
>
> **Failure cases and limitations**
>
> 1. Inherited annotation errors: When the weak label itself is wrong (mislabeled coarse regions), our weak-label-aware mask selection faithfully propagates the error. See Fig. 10 in the supplement, where there is a mismatch between label annotation in coarse vs fine annotation.
>
> 2. Stuff classes with ambiguous boundaries (vegetation, sky): SAM masks do not align well with ground truth annotations of stuff classes. Since stuff classes do not have defined shape and can cover big portion of the scene, sometimes SAM masks do not cover it entirely (see Fig. 13 in the supplement).
>
> 3. Thin structures (pole, fence, traffic sign): SAM tends to merge thin structures with their background (see Fig. 13 in the supplement).
>
> We have added a ``Limitations and Failure Cases'' discussion in the paper in Sec. A.2.
>
> ---
>
> **6. Stronger SAM baseline using the automatic mask generator**
>
> We thank the reviewer for this suggestion - it indeed makes our contribution claim stronger. Following the reviewer's suggestion, we ran an additional, stronger baseline based on SAM's automatic mask generator (AMG) similar to [A]: dense grid prompting produces class-agnostic masks over the whole image, each AMG mask is then assigned the semantic class of the weak label with which it has maximal overlap (masks with no overlapping weak label are left unlabeled/ignored), and the segmentation network is trained on the resulting labels under the identical training protocol. Notably, SAM-AMG improves over the naïve baseline for scribble and coarse labels but not for point labels. For points, the single annotated location is already an ideal SAM prompt, so naïve prompting yields a clean mask; AMG discards this location and instead assigns grid-generated masks by weak-label overlap resulting in poorer mask. For scribble and coarse labels the naïve prompt is itself unreliable, whereas AMG's overlap-based assignment helps to obtain better masks.
>
> Table 1: Comparison on PASCAL VOC 2012 using scribble and point labels with DeepLab segmentation methods.
>
> | Method | DeepLab | Scribble mIoU | Scribble WvF | Point mIoU | Point WvF |
> |---|---|---|---|---|---|
> | Pan et al. (Pan et al., 2021) | V2 | 74.6 | 96.0 | – | – |
> | SPML (Ke et al., 2021) | V2 | 74.2 | 95.5 | – | – |
> | A2GNN (Zhang et al., 2021) | V2 | 74.3 | 95.5 | 66.8 | 85.9 |
> | Supervised | V3+ | 69.7 | 87.6 | 61.6 | 77.4 |
> | FixMatch (Sohn et al., 2020) | V3+ | 71.3 | 89.6 | 68.4 | 86.0 |
> | DBFNet (Wu et al., 2022) | V3+ | 72.5 | 91.2 | 66.8 | 84.0 |
> | TEL (Liang et al., 2022) | V3+ | 75.8 | 96.9 | 63.3 | 86.0 |
> | AGMM (Wu et al., 2023) | V3+ | 76.4 | 96.1 | 69.6 | 87.5 |
> | AGMM* (Wu et al., 2023) | V3+ | 76.7 | 96.4 | 68.7 | 86.4 |
> | SAM baseline | V3+ | 38.4 | 48.3 | 71.1 | 89.4 |
> | **SAM-AMG (chen et al., 2023)** | **V3+** | **76.2** | **95.8** | **67.2** | **84.5** |
> | Ours | V3+ | 78.1 | 98.2 | 75.2 | 94.6 |
>
>
> Table 2:
>
> | Annotation | Method | SegFormer Cityscapes | SegFormer ADE20k | SegFormer PASCAL | DeepLabV3+ Cityscapes | DeepLabV3+ ADE20k |
> |---|---|---|---|---|---|---|
> | Point | Supervised | 51.8 | 26.5 | 61.5 | 52.2 | 29.5 |
> | | FixMatch (Sohn et al., 2020) | 53.1 | 27.2 | 63.6 | 52.9 | 29.8 |
> | | AGMM* (Wu et al., 2023) | 53.4 | – | 56.6 | 52.4 | – |
> | | SAM baseline | 52.1 | 30.9 | 60.9 | 53.9 | 34.9 |
> | | **SAM-AMG (chen et al., 2023)** | **47.0** | **26.6** | **60.0** | **53.8** | **30.3** |
> | | Ours | 58.5 | 32.5 | 67.4 | 61.8 | 36.3 |
> | Scribble | Supervised | 62.6 | 35.5 | 66.4 | 63.8 | 39.6 |
> | | FixMatch (Sohn et al., 2020) | 64.2 | 35.8 | 67.1 | 64.8 | 40.4 |
> | | AGMM* (Wu et al., 2023) | 56.7 | – | 68.1 | 67.4 | – |
> | | SAM baseline | 27.0 | 11.1 | 35.5 | 28.1 | 11.7 |
> | | **SAM-AMG (chen et al., 2023)** | **60.9** | **32.3** | **67.7** | **70.2** | **40.6** |
> | | Ours | 71.6 | 37.0 | 71.5 | 75.3 | 43.3 |
> | Coarse | Supervised | 65.2 | 35.7 | 62.8 | 67.9 | 40.9 |
> | | FixMatch (Sohn et al., 2020) | 66.7 | 36.7 | 63.6 | 68.9 | 41.3 |
> | | AGMM* (Wu et al., 2023) | 57.1 | – | 58.6 | 70.6 | – |
> | | SAM baseline | 43.1 | 24.2 | 41.4 | 48.5 | 27.0 |
> | | **SAM-AMG (chen et al., 2023)** | **61.9** | **32.1** | **66.5** | **69.9** | **39.5** |
> | | Ours | 69.9 | 37.3 | 70.2 | 72.5 | 43.2 |
>
> ---

---

> ### Author Response · Authors · 2026-07-05
>
> **7. Discussion/comparison with [A] and [B]**
>
> We have added both works to the Related Work section (Sec. 2); thank you for the pointers.
>
> [A] SEPL - SAM Enhanced Pseudo Labels for WSSS (arXiv'23). This work is in fact already cited in our submission (Chen et al., 2023a, Sec. 2); SEPL operates in the image-level-label WSSS setting: it uses SAM's automatic mask generator to post-process CAM-derived pseudo labels on PASCAL. It thus differs from SeSAM on all three axes we study - supervision type (image-level vs. fine-grained point/scribble/coarse), prompting (annotation-agnostic grid vs. annotation-aware skeleton sampling), and benchmark difficulty (PASCAL vs. also Cityscapes/ADE20k). As discussed above and in Sec. A.2, CAM-seeded approaches degrade severely on real-world multi-class scenes; the new AMG baseline additionally serves as a controlled proxy for the SEPL-style strategy within our setting.
>
> [B] ESC-Net - Effective SAM Combination for Open-Vocabulary Semantic Segmentation (CVPR'25). ESC-Net addresses a different problem: open-vocabulary semantic segmentation, where class names are given as text and the model leverages CLIP-style vision-language alignment together with SAM's decoder, generating pseudo prompts from image-text correlation rather than from spatial weak annotations. It assumes a pretrained vision-language model and does not study weakly supervised segmentation. We have added it in Sec. 2 as a complementary direction (text-driven semantics vs. weak-label-driven semantics).
>
> ---
>
> We believe these additions: two new baselines, a stronger AMG-based SAM baseline, clarified motivation and method exposition, expanded related work, and a dedicated limitations/failure-case analysis addresses the reviewer's concerns, and we thank the reviewer again for helping us substantially strengthen the paper.

---

### Review · Reviewer_GnCL · 2026-05-15

**Summary Of Contributions:**

In this paper, the authors study the problem of weakly-supervised semantic segmentation. To be specific, they focus on taking the SAM foundation model, which can perform instance segmentation very well, and adapt it to perform semantic segmentation. They use SAM to obtain weak-supervision/pseudo labels and refine them to obtain better semantic segmentation masks. The propose method is evaluated on several datasets and compared against several baselines.

**Audience:**

Yes

**Audience Explanation:**

People working on semantic segmentation might be interested in this paper.

**Broader Impact Concerns:**

No need for a broader impact statement.

**Claims And Evidence:**

No

**Claims Explanation:**

The experimental evaluation is weak, including only outdated baselines.

**Requested Changes:**

Strengths:

+ The use of SAM for semantic segmentation is an important problem.
+ Strong results are reported over the baselines.


Weaknesses:

1. I find the overall approach an engineering effort rather than incorporating any interesting new scientific insight or finding.

2. The methodology section needs improvements.

2.1. "Specifically, we first obtain an initial coarse mask by prompting SAM with the available weak labels." => These available weak labels refer to "point, scribble, and coarse annotations" but it is not clear how they are obtained.

2.2. "Each class mask is decomposed into a set of Kc disjoint instance masks using the connected component operator" => How the connected components are obtained is very crucial. For self-sufficiency, the paper should explain the specific method in detail.

2.3. Please introduce an algorithm that clearly explains the inputs, the steps and the outputs.

2.4. "(1) most of the points should be positioned near the central line of the mask to capture central details and connection between the di!erent sub parts of the object; (2) some points should be close to the boundaries to refine edges; and (3) all K sampled points should be uniformly distributed throughout the region." => These are important details for which you have some justifying visualizations in the Appendix. To be able to better follow the design choices, I suggest that you include these visualizations in the main paper.

3. Experimental evaluation is not convincing.

3.1. Yes, significant gains are reported over some baselines. However, the chosen baselines are from 2020, 2023 etc.

3.2. There should be comparisons with other weakly-supervised semantic segmentation studies.


Minor comments:
- "class ’person’" => "class `person’". The direction of the opening quote is important.
- All equations should be numbered. Please follow the following guide:
https://wp.optics.arizona.edu/kupinski/wp-content/uploads/sites/91/2023/05/MerminEquations.pdf

---

> ### Author Response · Authors · 2026-07-05
> **Thanks for the review**
>
> We thank Reviewer GnCL and are glad that the reviewer found the problem important and our results strong over the baselines. To directly address the baseline-recency concern, we add two recent (ICML'24, CVPR'25) fine-grained WSSS baselines and a stronger grid-prompt SAM baseline (Tab. 1 and Tab. 2, Reviewer 3 feedback). We address each point below.
>
> ---
>
> **1. Overall approach an engineering effort**
>
> We disagree that the work is purely engineering. We first discuss the scope as a contribution and then provide several important findings -
>
> **Scope (contribution in itself).** To our knowledge this is the first WSSS study spanning all three fine-grained label types (point, scribble, coarse) and three datasets (PASCAL, Cityscapes, ADE20k), with both CNN and transformer backbones (Tabs. 1 and 2). Prior fine-grained work (Pan et al., 2021; Ke et al., 2021; Zhang et al., 2021; Wu et al., 2022; 2023; Liang et al., 2022) treats point or scribble in isolation, almost always on PASCAL, and never coarse. Because there is so little prior work in this direction, we intend it to serve as a benchmark the community can build on facilitating follow up works.
>
> **Findings:**
>
> 1. **Instance priors help, but adaptation is necessary, not optional.** Naïve SAM degrades WSSS sharply (−44.6 mIoU, Cityscapes/scribble, Tab. 2; also confirmed by the new grid-prompt baseline), refuting the assumption that a strong foundation model trivially can work with weak supervision. We isolate the causes: instance/class mismatch, prompt-placement sensitivity, granularity ambiguity (Fig. 2), and ablate each (Tabs. 3 and 4).
>
> 2. **Confidence does not identify the correct granularity.** SAM generates masks at multiple granularities with associated confidence scores; the highest-scoring one is not the best match to the target class (−0.9 mIoU vs. our weak-label-aware selection, Tab. 4b). Selecting by intersection with the weak annotation is more reliable.
>
> 3. **First annotation-budget analysis.** Fig. 5 plots mIoU vs. actual annotation hours across all label types and dataset scales. In particular. scribbles reach 94% of full supervision at 2% of the budget, and matches it at 6%, turning "which weak label to collect" into a quantified decision, helping practitioners.
>
> 4. **Prompting geometry matters.** Effective point prompting is globally uniform but locally skeleton-biased; center-only or boundary-only, and confidence-based sampling each fail characteristically (Tab. 4a) generating non optimal masks.
>
> 5. **Cheaper annotation can be more reliable.** Scribbles outperform manual coarse labels, which we trace back to coarse-label noise (wall/fence/terrain systematically mislabeled, Tab. 9, Fig. 10) and higher object label recall (31.5 vs. 27.5 objects/image).
>
> ---

---

> ### Author Response · Authors · 2026-07-05
>
> **2.1 How are the weak labels (point, scribble, coarse) obtained?**
>
> Sorry for the confusion. We provide the details on the weak labels in Sec. 4 ("Datasets and metrics used"). In particular, the weak labels are obtained following prior works:
>
> Points: For PASCAL we use standard point annotation following [1], for Cityscapes and ADE20k we follow [2] to obtain the point labels.
>
> Scribbles: The public scribble annotations from [3] for PASCAL and the Scribbles-for-All benchmark [4] for Cityscapes/ADE20k.
>
> Coarse: Cityscapes' official manually annotated coarse set [5]; for ADE20k and PASCAL we obtain course masks following [6].
>
> **How are the initial coarse masks obtained?**
>
> As described in Sec. 3.2 ("Types of Weak Supervision"), point and scribble labels undergo a sparse-to-coarse conversion. For point labels, the labeled point is passed directly to SAM as a prompt; for scribbles, we sample 5 points along the scribble (using the same sampling strategy as SeSAM) and prompt SAM with them. Among the three masks SAM returns, we select the smallest-area one, favoring precision, as the initial coarse mask. The SeSAM pipeline then progressively refines and expands this initial mask during training.
>
> [1] Bearman et al., What's the Point: Semantic Segmentation with Point Supervision, ECCV 2016
> [2] Paul et al., Domain Adaptive Semantic Segmentation Using Weak Labels, ECCV 20
> [3] Lin et al., Scribble-supervised convolutional networks for semantic segmentation, CVPR 2016
> [4] Boettcher et al., Scribbles for all: Benchmarking scribble supervised segmentation across datasets, NeurIPS 2024
> [5] Cordts et al., The cityscapes dataset for semantic urban scene understanding, CVPR 2016
> [6] Das et al., Urban scene semantic segmentation with low-cost coarse annotation, WACV 2023
>
> ---
>
> **2.2 How to obtain K_c instances from a mask?**
>
> We obtain connected components using the standard labeling algorithm implemented in `scipy.ndimage.label`. Given the binary mask M⁽ᶜ⁾ for class c, the algorithm assigns a unique integer ID to each set of foreground pixels that are mutually connected, which we treat as individual instances. We use the default 4-connectivity structuring element. This function is deterministic and hyperparameter-free. We have updated the discussion with the library detail, and connectivity choice explicitly in Sec. 3.2.
>
> ---
>
> **2.3 Introduce an algorithm box**
>
> Thanks for the suggestion, we have added an algorithm box in the paper (see Alg. 1 in section A.2).
>
> ---
>
> **2.4 Move sampling strategy visualization to the main paper.**
>
> Thanks for the suggestion, we have moved the visualization to the main paper (Fig. 4, Sec. 3.2). Additionally, as requested by reviewer Mx9m, we have also extended it with step by step visualization showing each step for the sampling strategy.
>
> ---

---

> ### Author Response · Authors · 2026-07-05
>
> **3.1 Experimental evaluation with baselines from 2020, 2023 etc**
>
> We thank the reviewer for pushing on this. We would first like to clarify the state of the field, and then report new comparisons.
>
> **State of the field:** Our work targets WSSS with fine-grained weak labels (points, scribbles, coarse masks). The large majority of recent WSSS literature, including most 2024–2025 SAM-based works, operates with image-level labels and is evaluated only on PASCAL (see the comprehensive list at https://github.com/pengtaojiang/awesome-weakly-supervised-semantic-segmentation-papers
>
> As we show in the supplement (Sec. A.2, Fig. 11), image-level methods do not transfer to real-world scenes: the strong image-label SAM-based method, SAM2CAM (CVPR'24), achieves only 5.8 mIoU on Cityscapes vs. 61.8 for SeSAM with point labels under the same DeepLabv3+ network, because frequent class co-occurrence (e.g., road+car in nearly every image) breaks CAM localization. This is why our comparisons focus on fine-grained-label methods, with AGMM as a strong baseline at submission time.
>
> **New comparisons.** Following an exhaustive search, we identified two recent fine-grained WSSS works and have added them as baselines:
>
> 1. PBFA: Scribble-Supervised Semantic Segmentation with Prototype-based Feature Augmentation (ICML'24). Augments scribble supervision by hallucinating features around class prototypes mined from the labeled scribble pixels. Like most prior fine-grained work, it is designed and evaluated for scribbles on PASCAL only, and uses no foundation-model priors. We compare on PASCAL and additionally re-implement and extend it to Cityscapes and ADE20k.
>
> 2. SSL-Potts: Soft Self-labeling and Potts Relaxations for Weakly-Supervised Segmentation (CVPR'25). Improves scribble-supervised training via soft pseudo-labels regularized with relaxed Potts/CRF objectives. It is likewise scribble-centric and prior-free. We compare directly on all three datasets.
>
> Neither prior method handles coarse annotations or the challenging Cityscapes/ADE20k benchmarks out of the box.
>
> | Method | PASCAL (Scribble) | Cityscapes (Scribble) | ADE20k (Scribble) |
> |---|---|---|---|
> | PBFA (ICML'24) | 76.2 | 67.5 | 40.0 |
> | SSL-Potts (CVPR'25) | 78.1 | 72.4 | 39.7 |
> | SeSAM (Ours) | 78.1 | 75.3 | 43.3 |
>
> SeSAM overall outperforms both recent baselines. We have added these comparisons to Tables 1-2 and the Related Work discussion in the revision.
>
> **Minor comments**
>
> - Quote direction ("class 'person'"): thank you for catching this. We have fixed this.
> - Equation numbering: Thanks for the suggestion. We have updated the draft with equation numbers.
>
> ---
>
> We believe these additions: two recent fine-grained baselines (PBFA, SSL-Potts) and a stronger AMG-based SAM baseline (see Tab. 1 and 2, and Rev 3 point), clarified contribution, clarified method (connected components, sampling goals, algorithm box), and moved sampling visualization to main paper, addresses the reviewer's concerns. We thank the reviewer again for helping us substantially strengthen the paper.

---

### Review · Reviewer_Jxsc · 2026-06-22

**Summary Of Contributions:**

**Summary**

This paper studies whether instance priors from SAM can improve weakly supervised semantic segmentation under point, scribble, and coarse-mask supervision. The authors argue that SAM cannot be directly used for semantic segmentation because it is instance-centric, prompt-sensitive, and produces masks at different granularities. To address these issues, the paper proposes SeSAM, a framework that decomposes class masks into connected components, samples point prompts from instance-level masks using a skeleton-aware strategy, selects SAM masks based on weak-label coverage/compatibility, and integrates the resulting SAM-derived labels into a semi-supervised teacher-student training framework together with weak annotations and pseudo-labels. The reported results show consistent improvements over supervised weak-label baselines, FixMatch, AGMM, and a naive SAM baseline.

**Strengths**

1. The paper clearly identifies why directly applying SAM to weakly supervised semantic segmentation is non-trivial: SAM is instance-based, semantic segmentation is class-based, prompt locations matter, and SAM mask granularity can be ambiguous. The three main design choices, instance separation, point sampling, and mask selection, are intuitive and well motivated.

2. Interesting and practical use of SAM as an instance prior.

3. Strong empirical evaluation and comprehensive detailed analysis.


**Weaknesses**

1. The current naive SAM baseline may not fully isolate the contribution of SeSAM’s refinement strategy. The current SAM baseline is useful, but it does not fully isolate the contribution of SeSAM’s label-refinement steps because SeSAM is trained within a semi-supervised framework that also uses pseudo-labels. A stronger comparison would keep the same semi-supervised pipeline but replace the refined SeSAM labels with naively generated SAM labels.

2. Failure cases and limitations are under-discussed.

3. It would be better to report standard deviations over multiple runs for the main results. Since the method involves stochastic training, pseudo-labeling, and point sampling for SAM prompts, reporting mean ± std would make the empirical gains more reliable and easier to interpret.

**Audience:**

Yes

**Audience Explanation:**

The paper addresses a timely and practical question: how to use a powerful promptable segmentation foundation model such as SAM when only weak labels are available.

**Claims And Evidence:**

Yes

**Claims Explanation:**

The experimental evidence broadly supports the main claims. The paper claims that SAM instance priors can improve weakly supervised semantic segmentation when properly adapted, and the results across PASCAL VOC, Cityscapes, and ADE20k support this claim. The comparison against weak-label supervised training, FixMatch, AGMM, and the naïve SAM baseline provides a convincing empirical case. The ablations further support the importance of instance separation, mask selection, prompt sampling, and refined-mask sampling.

**Requested Changes:**

1. Add a matched “naive SAM + semi-supervised training” baseline (weakness 1).

2. Discuss failure cases and limitations.

3. Report std. in main experiments.

---

> ### Author Response · Authors · 2026-07-05
>
> We thank Reviewer Jxsc for the positive review, and are glad that the reviewer found our motivation clear, our three design choices well motivated, and our empirical evaluation strong. We address the three requested changes below.
>
> ---
>
> **1. Comparison with better SAM baseline.**
>
> Thank you for suggesting this experiment. We performed this experiment and updated SAM baseline to include semi-supervised pipeline. The results are presented below. Overall the performance with additional semi-supervised pipeline improves over naive baseline but still is lower compared to our approach. We have updated the SAM-baseline in our paper with these new results.
>
> Table 1: Comparison on PASCAL VOC 2012 using scribble and point labels with DeepLab segmentation methods.
>
> | Method | DeepLab | Scribble mIoU | Scribble WvF | Point mIoU | Point WvF |
> |---|---|---|---|---|---|
> | Pan et al. (Pan et al., 2021) | V2 | 74.6 | 96.0 | – | – |
> | SPML (Ke et al., 2021) | V2 | 74.2 | 95.5 | – | – |
> | A2GNN (Zhang et al., 2021) | V2 | 74.3 | 95.5 | 66.8 | 85.9 |
> | Supervised | V3+ | 69.7 | 87.6 | 61.6 | 77.4 |
> | FixMatch (Sohn et al., 2020) | V3+ | 71.3 | 89.6 | 68.4 | 86.0 |
> | DBFNet (Wu et al., 2022) | V3+ | 72.5 | 91.2 | 66.8 | 84.0 |
> | TEL (Liang et al., 2022) | V3+ | 75.8 | 96.9 | 63.3 | 86.0 |
> | AGMM (Wu et al., 2023) | V3+ | 76.4 | 96.1 | 69.6 | 87.5 |
> | AGMM* (Wu et al., 2023) | V3+ | 76.7 | 96.4 | 68.7 | 86.4 |
> | SAM baseline | V3+ | 38.4 | 48.3 | 70.3 | 88.4 |
> | **SAM baseline (updated)** | **V3+** | **38.4** | **48.3** | **71.1** | **89.4** |
> | Ours | V3+ | 78.1 | 98.2 | 75.2 | 94.6 |
>
> Table 2: Comparison across datasets (Cityscapes, ADE20k, PASCAL VOC 2012), label types (point, scribble, coarse) and architectures (DeepLabV3+ and SegFormer)
>
> | Annotation | Method | SegFormer Cityscapes | SegFormer ADE20k | SegFormer PASCAL | DeepLabV3+ Cityscapes | DeepLabV3+ ADE20k |
> |---|---|---|---|---|---|---|
> | Point | Supervised | 51.8 | 26.5 | 61.5 | 52.2 | 29.5 |
> | | FixMatch (Sohn et al., 2020) | 53.1 | 27.2 | 63.6 | 52.9 | 29.8 |
> | | AGMM* (Wu et al., 2023) | 53.4 | – | 56.6 | 52.4 | – |
> | | SAM baseline | 51.0 | 29.3 | 59.0 | 50.0 | 35.1 |
> | | **SAM baseline (updated)** | **52.1** | **30.9** | **60.9** | **53.9** | **34.9** |
> | | Ours | 58.5 | 32.5 | 67.4 | 61.8 | 36.3 |
> | Scribble | Supervised | 62.6 | 35.5 | 66.4 | 63.8 | 39.6 |
> | | FixMatch (Sohn et al., 2020) | 64.2 | 35.8 | 67.1 | 64.8 | 40.4 |
> | | AGMM* (Wu et al., 2023) | 56.7 | – | 68.1 | 67.4 | – |
> | | SAM baseline | 24.2 | 10.6 | 34.8 | 26.3 | 11.2 |
> | | **SAM baseline (updated)** | **27.0** | **11.1** | **35.5** | **28.1** | **11.7** |
> | | Ours | 71.6 | 37.0 | 71.5 | 75.3 | 43.3 |
> | Coarse | Supervised | 65.2 | 35.7 | 62.8 | 67.9 | 40.9 |
> | | FixMatch (Sohn et al., 2020) | 66.7 | 36.7 | 63.6 | 68.9 | 41.3 |
> | | AGMM* (Wu et al., 2023) | 57.1 | – | 58.6 | 70.6 | – |
> | | SAM baseline | 42.4 | 23.5 | 41.3 | 47.9 | 26.9 |
> | | **SAM baseline (updated)** | **43.1** | **24.2** | **41.4** | **48.5** | **27.0** |
> | | Ours | 69.9 | 37.3 | 70.2 | 72.5 | 43.2 |
>
> ---
>
> **2. Failure cases and limitations**
>
> 1. Inherited annotation errors: When the weak label itself is wrong (mislabeled coarse regions), our weak-label-aware mask selection faithfully propagates the error. See Fig. 10 in the supplement, where there is a mismatch between label annotation in coarse vs fine annotation.
>
> 2. Stuff classes with ambiguous boundaries (vegetation, sky): SAM masks do not align well with ground truth annotations of stuff classes. Since stuff classes do not have defined shape and can cover big portion of the scene, sometimes SAM masks do not cover it entirely (see Fig. 13 in the supplement).
>
> 3. Thin structures (pole, fence, traffic sign): SAM tends to merge thin structures with their background (see Fig. 13 in the supplement).
>
> We have added a ``Limitations and Failure Cases'' discussion in the paper in Sec. A.2.
>
> ---

---

> ### Author Response · Authors · 2026-07-05
>
> **3. Mean and std deviation in main results**
>
> Thank you for this suggestion. We conducted each experiment with three different runs and report the mean and standard deviation in the below tables. We have updated these results in the main paper (Tab.1 and 2).
>
> Table 1: Comparison on PASCAL VOC 2012 using scribble and point labels with DeepLab segmentation methods.
>
> | Method | DeepLab | Scribble mIoU | Scribble WvF | Point mIoU | Point WvF |
> |---|---|---|---|---|---|
> | Pan et al. (Pan et al., 2021) | V2 | 74.6 | 96.0 | – | – |
> | SPML (Ke et al., 2021) | V2 | 74.2 | 95.5 | – | – |
> | A2GNN (Zhang et al., 2021) | V2 | 74.3 | 95.5 | 66.8 | 85.9 |
> | Supervised | V3+ | 69.7 | 87.6 | 61.6 | 77.4 |
> | FixMatch (Sohn et al., 2020) | V3+ | 71.3 | 89.6 | 68.4 | 86.0 |
> | DBFNet (Wu et al., 2022) | V3+ | 72.5 | 91.2 | 66.8 | 84.0 |
> | TEL (Liang et al., 2022) | V3+ | 75.8 | 96.9 | 63.3 | 86.0 |
> | AGMM (Wu et al., 2023) | V3+ | 76.4 | 96.1 | 69.6 | 87.5 |
> | AGMM* (Wu et al., 2023) | V3+ | 76.7 | 96.4 | 68.7 | 86.4 |
> | SAM baseline | V3+ | 38.4 | 48.3 | 71.1 | 89.4 |
> | **Ours** | **V3+** | **78.1±0.2** | **98.2** | **75.2±0.1** | **94.6** |
>
>
>
> Table 2:
>
> | Annotation | Method | SegFormer Cityscapes | SegFormer ADE20k | SegFormer PASCAL | DeepLabV3+ Cityscapes | DeepLabV3+ ADE20k |
> |---|---|---|---|---|---|---|
> | Point | Supervised | 51.8 | 26.5 | 61.5 | 52.2 | 29.5 |
> | | FixMatch (Sohn et al., 2020) | 53.1 | 27.2 | 63.6 | 52.9 | 29.8 |
> | | AGMM* (Wu et al., 2023) | 53.4 | – | 56.6 | 52.4 | – |
> | | SAM baseline | 52.1 | 30.9 | 60.9 | 53.9 | 34.9 |
> | | **Ours** | **58.5±0.3** | **32.5±0.1** | **67.4±0.3** | **61.8±0.1** | **36.3±0.1** |
> | Scribble | Supervised | 62.6 | 35.5 | 66.4 | 63.8 | 39.6 |
> | | FixMatch (Sohn et al., 2020) | 64.2 | 35.8 | 67.1 | 64.8 | 40.4 |
> | | AGMM* (Wu et al., 2023) | 56.7 | – | 68.1 | 67.4 | – |
> | | SAM baseline | 27.0 | 11.1 | 35.5 | 28.1 | 11.7 |
> | | **Ours** | **71.6±0.2** | **37.0±0.1** | **71.5±0.3** | **75.3±0.1** | **43.3±0.2** |
> | Coarse | Supervised | 65.2 | 35.7 | 62.8 | 67.9 | 40.9 |
> | | FixMatch (Sohn et al., 2020) | 66.7 | 36.7 | 63.6 | 68.9 | 41.3 |
> | | AGMM* (Wu et al., 2023) | 57.1 | – | 58.6 | 70.6 | – |
> | | SAM baseline | 43.1 | 24.2 | 41.4 | 48.5 | 27.0 |
> | | **Ours** | **69.9±0.1** | **37.3±0.1** | **70.2±0.2** | **72.5±0.3** | **43.2±0.1** |
>
> ---
>
> We believe these additions: a matched semi-supervised SAM baseline, a dedicated limitations and failure-case analysis, and mean ± std over three runs, fully address the reviewer's requested changes, and we thank the reviewer again for helping us strengthen the paper.

---

### Author Response · Authors · 2026-07-05
**Summary of Revisions**

**Summary of Revisions**

We thank all reviewers for their constructive feedback. We are encouraged that the reviewers found the problem timely and of broad interest (Jxsc, Mx9m), the use of SAM as an instance prior important and practical (GnCL, Jxsc), our design choices well-motivated (Jxsc), and our evaluation strong (GnCL, Jxsc). We revised the paper along three directions.

**Baselines and evaluation**
- Added two recent fine-grained baselines, PBFA (ICML'24) and SSL-Potts (CVPR'25) (Tables 1 and 2, Sec. 2, Sec 4.1) (GnCL 3, Mx9m 3).
- Added a stronger grid-prompt baseline, SAM-AMG (Tables 1 and 2, Sec. 4.1) (Mx9m 6).
- Updated the naive SAM baseline to run with semi-supervised pipeline, isolating our refinement (Tables 1 and 2, Sec. 4.1) (Jxsc 1).
- Reported mean ± std for main results (Tables 1 and 2) (Jxsc 3).

**Methodology and clarity**
- Added connected-component details: `scipy.ndimage.label`, 4-connectivity, deterministic (Sec. 3.2) (GnCL 2.2).
- Clarified the sampling goals as globally uniform but locally skeleton-biased (Sec. 3.2) (GnCL 2.4, Mx9m 2).
- Moved the skeleton-sampling visualization to the main paper, extended step-by-step (Fig. 4) (GnCL 2.4, Mx9m 2).
- Added an algorithm box (Algorithm 1, Appendix A.2) (GnCL 2.3).

**Limitations and related work**
- Added a "Failure cases and limitations" discussion (Sec. A.2, Figs. 10, 13) (Jxsc 2, Mx9m 5).
- Expanded related work with SEPL (arxiv 23), ESC-Net (CVPR 25), PBFA (ICML 24), and SSL-Potts (CVPR 25) (Sec. 2) (Mx9m 7).

We believe these revisions address the reviewers' concerns and strengthen the paper, and thank the reviewers again for their help.